# Phylogeny of *PmCCD* Gene Family and Expression Analysis of Flower Coloration and Stress Response in *Prunus mume*

**DOI:** 10.3390/ijms241813950

**Published:** 2023-09-11

**Authors:** Aiqin Ding, Fei Bao, Wenhui Cheng, Tangren Cheng, Qixiang Zhang

**Affiliations:** 1Beijing Key Laboratory of Ornamental Plants Germplasm Innovation and Molecular Breeding, National Engineering Research Center for Floriculture, School of Landscape Architecture, Beijing Forestry University, Beijing 100083, China; dingaiqin0728@163.com (A.D.); cwh13452454084@bjfu.edu.cn (W.C.); chengtangren@163.com (T.C.); 2Beijing Laboratory of Urban and Rural Ecological Environment, Engineering Research Center of Landscape Environment of Ministry of Education, School of Landscape Architecture, Beijing Forestry University, Beijing 100083, China; 3Key Laboratory of Genetics and Breeding in Forest Trees and Ornamental Plants of Ministry of Education, School of Landscape Architecture, Beijing Forestry University, Beijing 100083, China

**Keywords:** *Prunus mume*, *PmCCDs*, flower coloration, stress response, expression pattern, protein interaction

## Abstract

The *CCD* gene family plays a crucial role in the cleavage of carotenoids, converting them into apocarotenoids. This process not only impacts the physiology and development of plants but also enhances their tolerance toward different stresses. However, the character of the *PmCCD* gene family and its role in ornamental woody *Prunus mume* remain unclear. Here, ten non-redundant *PmCCD* genes were identified from the *P. mume* genome, and their physicochemical characteristics were predicted. According to the phylogenetic tree, PmCCD proteins were classified into six subfamilies: CCD1, CCD4, CCD7, CCD8, NCED and CCD-like. The same subfamily possessed similar gene structural patterns and numbers of conserved motifs. Ten *PmCCD* genes were concentrated on three chromosomes. *PmCCD* genes exhibited interspecific collinearity with *P. armeniaca* and *P. persica*. Additionally, *PmCCD* genes had obvious specificity in different tissues and varieties. Compared with white-flowered ‘ZLE’, *PmCCD1* and *PmCCD4* genes were low-expressed in ‘HJH’ with yellow petals, which suggested *PmCCD1* and *PmCCD4* might be related to the formation of yellow flowers in *P. mume*. Nine *PmCCD* genes could respond to NaCl or PEG treatments. These genes might play a crucial role in salt and drought resistance in *P. mume*. Moreover, PmVAR3 and PmSAT3/5 interacted with PmCCD4 protein in yeast and tobacco leaf cells. This study laid a foundation for exploring the role of the *PmCCD* gene family in flower coloration and stress response in *P. mume*.

## 1. Introduction

Carotenoids are a class of C40 terpenoid compounds and their derivatives, mainly composed of eight isoprene units. In plants, carotenoids play a role in photosynthesis assistance and photoprotection and the biosynthesis of plant hormones abscisic acid and strigolactone [1,2,3]. Moreover, they enable the flowers, fruits and other organs of higher plants to present a variety of brilliant colors and aromas, thus attracting birds and insects to participate in plant pollination and seed dispersal [4].

Carotenoid cleavage dioxygenases (CCDs) can oxidatively cleave carotenoids at one or both ends of the molecule to produce different apocarotenoids, which participate in plant growth, development and stress response [5]. Tan et al. [6] identified the first carotenoid cleavage dioxygenases viviparous 14 (VP14) in maize mutants. Subsequently, *CCD* genes were identified in multiple species. In *Arabidopsis thaliana*, *CCD* genes were divided into two categories: carotenoid cleavage dioxygenases (CCD) and 9-cis-epoxycarotenoid cleavage dioxygenases (NCEDs), according to whether the substrate of its catalytic cleavage undergoes epoxidation. The CCD subfamily mainly included CCD1, CCD4, CCD7 and CCD8.The NCED subfamily included NCED2, NCED3, NCED5, NCED6 and NCED9 [7]. Since then, the homologs found in other plants have been named in the classification of *Arabidopsis* manner. In addition, a novel group known as CCD-like (CCDL) has been identified in *Oryza sativa*, *Sorghum bicolor*, *Solanum lycopersicum* and *Malus pumila* [8,9,10].

The essential physiological functions of carotenoid lysates in plants have attracted significant attention to the lyases and their products involved in carotenoid metabolism. The main functional principle of *CCD* genes is to specifically cleave the double bonds on different carotenoid substrates to generate a variety of cleavage products, thus playing different biological functions. The substrates of *CCD1* genes have universality, which can preferentially and symmetrically cut the double bond at the 9,10 (9′, 10′) position of the carotenoid to produce flavor and aromatic substances as well as aldehydes or ketones. Simultaneously, it can also cleave double bonds at positions 5,6 (5′, 6′) and 7,8 (7′, 8′). In *Arabidopsis*, *S. lycopersicum*, *Cucumis melo*, *Petunia hybrida*, *Laurus nobilis*, *Fragaria ananassa* and *Lycium chinense*, the involvement of *CCD1* genes has been reported in the biosynthesis of volatile aromas, such as β-ionone [11,12,13,14,15,16,17]. *CCD4* genes play a crucial role in plant color formation and aroma production. In *Chrysanthemum morifolium*, inhibiting the expression of the *CmCCD4a* gene through RNAi could lead to a mutation from white to yellow flowers [18]. Suppressing the *CCD4* gene in potato caused the carotenoid in the tubers to increase and appear yellow [19]. Differential expression of the *BnCCD4* gene led to the emergence of yellow and white *Brassica napus*, and *BnCCD4* could cleave α-carotene as a substrate to produce α- ionone [20]. In addition, a unique cleavage activity of *CitCCD4* was found in *citrus*, which could cleave at the 7, 8 (7′, 8′) double-bond sites β-cryptoxanthin or zeaxanthin to produce orange-red apocarotenoid β-citraurin [21]. Studies have revealed that plant *CCD1* and *CCD4* also actively participate in responses to heat, drought, salt and other stresses [22]. For example, overexpression of *CsCCD4b* enhanced the stress tolerance to salt, dehydration and oxidation in *Arabidopsis* [23]. In higher plants, CCD7 and CCD8 produced the signaling substance strigolactone by cleaving β-carotene at the 9′, 10′ double-bond sites [24]. Strigolactone has been implicated in governing the development of lateral roots and branches in plants, as well as response to stress [25]. Ectopic expression of *CpCCD7* and *CpCCD8* restored the branching phenotype of *Arabidopsis* mutants *max3-9* and *max4-1*, respectively [26]. Under phosphorus deficiency, *ZmCCD7* upregulated the gene expression [27]. Similarly, in *Poplar* trees subjected to H_2_O_2_, drought and salt stress, the *CCD8* gene actively responds to stress stimuli [28].

*NCEDs* can oxidatively cleave violaxanthin and neoxanthin with an epoxy structure at the 11,12 double-bond sites to generate xanthoxin, which represents the initial step of ABA biosynthesis in plants [29]. The *NCED* gene can directly affect plant growth, development and stress response by regulating ABA synthesis. Currently, *NCED3* has been proven to be involved in abiotic stress responses in multiple species [30,31,32,33]. In rice, *OsNCED1* and *OsNCED2* regulated the ABA levels in response to drought conditions, thereby enhancing drought tolerance [34,35]. Overexpression of *OsNCED5* enhanced the salt and water stress tolerance of rice leaves and accelerated leaf senescence. *OsNCED5* also altered plant size and leaf morphology and delayed seed germination and flowering time in *Arabidopsis* [36]. In addition, overexpression of *CrNCED1* in citrus improved tolerance to multiple abiotic stresses [37]. In *Stylosanthes guianensis*, drought stress induced the expression of *SgNCED1* and the accumulation of ABA [38]. Additionally, the expression of the *CstNCED* gene played a vital role in regulating ABA levels during style senescence, corm dormancy and drought stress in *Crocus sativus* [39].

*P. mume* is a famous ornamental tree, rich in flower color, elegant flower fragrance and tree shape. *CCD* genes have a major effect on flower color and fragrance formation, the morphological construction of growth and development and stress response in plants [5]. Therefore, it is significant to identify *CCD* gene family members in *P. mume*. Here, we identified 10 *PmCCD* genes and performed bioinformatics analysis, including gene physicochemical characteristics, a phylogenetic tree, chromosome distribution, promoters, collinearity and protein interactions. The expression profiles of *PmCCD* genes in various tissues, varieties of yellow and white flowers and stresses (NaCl or PEG treatments) were analyzed by qRT-PCR. The interacting proteins of *PmCCD4* were verified by using a yeast two-hybrid system and luciferase complementation experiment. In summary, this study laid the foundation for the functional research of the *PmCCD* genes and amplified candidate genes for the breeding of *P. mume*.

## 2. Results

### 2.1. Identification of CCD Gene Family Members in P. mume

To obtain HMM for the REP65 domain (PF03055), the Pfam database was utilized. HMMER3 software was applied to search for *CCD* genes in the *P. mume* genome with E-value ≤ 10^−5^. Finally, 10 *PmCCD* genes were identified and named following the *Arabidopsis* nomenclature.

The physicochemical properties and secondary structures of 10 PmCCD proteins were analyzed (Appendix A). The PmCCD proteins exhibited a length ranging from 247 to 622 amino acids, with 80% of the proteins having 500–600 aa. The PmCCD1-like-c protein was the smallest, while PmNCED3 encoded the most amino acids. Additionally, three members of the NCED subfamily encoded more than 600 aa. The protein molecular weight varied between 27.71 kDa (PmCCD1-like-c) and 69.06 kDa (PmCCD1-like-a). For isoelectric points, PmCCD1-like-b and PmNCED6 were 7.06 and 7.92, implying neutrality. The isoelectric points of the other six proteins were less than 7, indicating that they were acidic. The instability index ranged from 28.96 (PmCCD1-like-a) to 46.09 (PmCCD7), with three unstable proteins and seven stable proteins. The grand average of hydropathicity was negative, showing that all PmCCD proteins were soluble. The aliphatic index was 64.7 to 88.24 and indicated that CCD proteins had good thermal stability. PmCCD1-like-c had the lowest total number of negatively charged residues (Asp + Glu) and total number of positively charged residues (Arg + Lys). The maximum number of (Asp + Glu) in PmCCD1-like-a was 76, while the maximum number of (Arg + Lys) in PmNCED6 was 67.

Except for PmCCD1 and PmCCD1-like-c proteins, chloroplast targeting peptides were present at the other eight PmCCDs sequences, indicating that eight PmCCDs might be localized in the chloroplast. Subcellular localization analysis of the PmCCD proteins revealed that the majority of these proteins were found to be localized within the chloroplasts, while PmCCD1 and PmCCD1-like-c were localized in the cytoplasm, which was consistent with the prediction of signal peptides.

Secondary structure prediction showed that all the family members consisted of an α-helix, extended strand, β-sheet and random coil. Out of them, the β-sheet was lower than 10%, followed by the α-helix. Except for PmCCD1-like-a and PmCCD1-like-b, all members were lower than 20%. However, the proportion of random coil was the highest, PmCCD1-like-b was 41.18%, and the remaining was more than 50%, indicating that the secondary structure of CCD family members was mainly random coil (Appendix A).

### 2.2. Phylogenetic Analysis and Protein Sequence Alignment

To investigate the systematic evolutionary relationships and potential functions of the PmCCDs, we generated a phylogenetic tree of CCD proteins from *P. mume*, *Prunus armeniaca*, *Prunus persica* and *Arabidopsis* using the maximum likelihood method (Figure 1). Based on the classification system of *Arabidopsis*, the PmCCD proteins were divided into six subfamilies: CCD1, CCD4, CCD7, CCD8, NCED and CCD-like. Obviously, there was no CCD-like subfamily in *Arabidopsis*, but the CCD-like subfamily has been confirmed in many species [10,40,41]. This suggested that CCD genes differed significantly in the evolutionary process of different species. The CCD members were unevenly distributed among the subfamilies. The CCD1, CCD4, CCD7 and CCD8 subfamilies each contained one member, while the NCED and CCD-like subfamilies had three members each. The number and distribution of CCD proteins in *P. armeniaca* and *P. persica* were consistent with *P. mume*. Additionally, we found PmCCD proteins shared a closer genetic distance with ParCCDs.

Next, protein multiple sequence alignment was performed on 10 identified CCD family members. Ten proteins possessed a conserved RPE65 domain (Figure 2C). Further analysis revealed that CCD proteins contained conserved histidine residues, essential for their enzymatic function (Appendix A). These findings indicated that PmCCD proteins had the potential to exhibit enzymatic activity.

### 2.3. Gene Structure and Conserved Motif

To gain further search structural characteristics of *PmCCD* genes, the exon and intron patterns were analyzed by using the phylogenetic tree (Figure 2A,C). The *PmCCD* family showed considerable variation in gene structure, with exon numbers ranging from 1 to 14 and intron numbers ranging from 0 to 13. Among them, *PmCCD1* contained the most exons and introns. On the other hand, the *NCED* subfamily lacked introns, while the *PmCCD4* subfamily had a relatively brief structure with one intron and two exons. Further analysis revealed three members; specifically, *PmCCD7*, *PmCCD1-like-a* and *PmCCD1-like-c* were short of the untranslated region (UTR).

Then, using the MEME website to predict the motifs of PmCCD proteins, a total of 15 motifs were obtained. Figure 2B illustrates that the quantity of motifs in the PmCCD family changed from 6 to 15. Interestingly, although the NCED subfamily had a relatively simple gene structure, it contained all 15 motifs. The number of motifs differed greatly in the PmCCD-like subfamily members. Concretely, PmCCD1-like-b and PmCCD1-like-c had only 6 motifs, while PmCCD1-like-a contained 13 motifs. Except for PmCCD1-like-c, all PmCCD proteins contained 4, 5, 6 and 9 motifs, demonstrating significant conservation across these four motifs. The motif diagrams of the four motif elements illustrated that each motif comprised fully conserved sites, and these motifs might be related to their common function (Figure 2D). Meanwhile, the motif composition further supported the phylogenetic tree branch, indicating that the PmCCD4 protein was most closely related to PmNCED proteins, while PmCCD7 and PmCCD8 proteins had the most distant relationship with PmNCED.

### 2.4. Chromosomal Distribution and Collinearity Analysis

According to the location information of the *PmCCD* family genes on the chromosome, a gene distribution map was generated with MG2C software (Figure 3). The results indicated that all 10 *PmCCD* genes were located on the chromosome and clustered on three chromosomes: chr2, chr3 and chr5. The chromosomal distribution of *PmCCD* genes was uneven, with six *PmCCD* genes (60%) located on chromosome 2. And chromosome 3 and chromosome 5 each included two *PmCCD* genes.

Intraspecific collinearity analysis was performed on the 10 *PmCCD* genes, and collinearity was absent. Neither tandem duplication nor segmental duplication made a contribution to the expansion of the *PmCCD* gene family. To explore the evolutionary relationship between the *PmCCD* genes and other *Prunus* plants, we constructed a collinearity map for *P. mume*, *P. armeniaca* and *P. persica* (Figure 4). As depicted in the map, *PmCCD* genes formed eight or nine collinear gene pairs with *P. armeniaca* or *P. persica*. Moreover, the Ka/Ks ratios between the collinear *CCD* gene pairs were performed to investigate the adaptive evolutionary relationships among three species (Appendix A). The results revealed that the Ka/Ks values between *P. mume* and *P. armeniaca* varied from 0.1 to 0.8, while those between *P. mume* and *P. persica* varied from 0.1 to 0.7. All Ka/Ks ratios were <1, indicating that genes had undergone varying degrees of purifying selection. Next, to estimate the divergence time of orthologous gene pairs, Ks values were employed. The divergence time between *P. mume* and *P. armeniaca* started at 0.47, increased to 102.47 Mya and concentrated at 0.47–1.62 Mya. In *P. mume* and *P. persica*, it began at 0.86 and increased to 44.55 Mya, and 1.31–2.44 Mya occurred in most collinear genes. This suggested the CCD orthologous genes might diverge earlier in *P. mume* and *P. armeniaca*.

### 2.5. Promoter Cis-Acting Element Analysis

To predict the potential regulatory mechanisms of *PmCCD* genes, an examination of the cis-acting elements in the promoter region was performed (Figure 5). Based on their functional relevance, the promoter regions were categorized. The results displayed that the *PmCCD* gene promoter contained light-responsive, hormone-responsive, biotic and abiotic stress and plant growth and development elements. Environmental factors and hormonal signals might exert complex regulatory effects on the expression of the *PmCCD* genes. There were 11 types of light-responsive elements presented in the *PmCCD* genes, with *PmCCD8* having the highest number (9 types). Furthermore, *PmCCD1* harbored the highest number of light-responsive elements with 69, while *PmNCED6* exhibited the lowest number with 42. The *PmCCD* genes contained 10 types of hormone-responsive elements, including abscisic acid, auxin, salicylic acid, jasmonic acid and gibberellin. The ABRE (ABA response element) was the most extensively distributed among these elements, as it was present in all genes. Among these, *PmNCED6* held the most ABRE elements. Notably, the promoter of *PmCCDs* contained 12 responsive elements to biotic and abiotic stresses, including the MYB element, MYC element, W box (WRKY binding element), LTRE (low-temperature-responsive element), etc. This suggested that transcription factors such as MYB, MYC and WRKY might bind to *PmCCD* genes, exerting significant regulatory functions in, for example, low-temperature, drought, salt and trauma response. Furthermore, plant growth and development are primarily focused on circadian rhythm, endosperm development and meristem development processes.

### 2.6. Expression Patterns of PmCCD Genes in Various Tissues and Different Varieties

To elucidate the biological function of the *PmCCD* genes, we analyzed the expression patterns in various tissues and varieties. The tissue-specific expression patterns of the *PmCCD* genes are presented in Figure 6A and Figure 7A. The heatmap shows that the tissue expression patterns of *PmCCD* genes were divided into three categories (Figure 6A and Appendix A). The class I, II and III genes were mainly highly expressed in roots, flower buds or stems, respectively. Next, expression analysis of the stems, leaves, petals, fruits, sepals, stamens and pistils of ‘HJH’ were analyzed with qRT-PCR (Figure 7A). *PmCCD1*, *PmCCD4*, *PmNCED3*, *PmNCED5*, *PmCCD1-like-b* and *PmCCD1-like-c* were expressed in all tissues. We predicted the extensive involvement of these genes in plant growth and development. *PmCCD1* and *PmCCD4* shared a similar expression pattern, exhibiting high expression levels in petals, while *PmCCD4* also displayed significantly high expression in leaves. *PmCCD7* was highly expressed in stems and pistils, whereas *PmCCD8* displayed high expression levels in stems and leaves. *PmNCED3*, *PmNCED5* and *PmNCED6* belonged to the *NCED* subfamily and were highly expressed in stamens, fruits and leaves, respectively. This indicated that genes within the same family gradually acquired different functions as genes evolved. Members of the *PmCCD1-like* subfamily exhibited stem-specific expression, with low expression levels in other tissues. *PmCCD1-like* genes probably had a specialized function in the stem.

The expression analysis of the *PmCCD* genes was carried out during the flowering process in *P. mume* ‘HJH’ with yellow flowers and *P. mume* ‘ZLE’ with white flowers using transcriptome sequencing (Figure 6B and Appendix A). The results indicated significant variation in the expression of the *PmCCD* genes across different varieties and flowering stages. Two distinct groups were formed based on the expression patterns of the *PmCCD* genes. Group I comprised the genes *PmNCED6*, *PmCCD7*, *PmCCD8*, *PmCCD1-like-a*, *PmCCD1-like-b* and *PmCCD1-like-c*. They exhibited negligible expression levels during the flowering stages in two varieties, consistent with the tissue specificity. The finding implied that group I genes were not involved in forming yellow flowers in *P. mume*. In addition, we observed distinct expression patterns in the *NCED* subfamily. *PmNCED3* and *PmNCED5* belonged to group II. *PmNCED3* was highly expressed in ‘HJH’-S2, while *PmNCED5* was highest in ‘ZLE’-S2. In group II, *PmCCD1* and *PmCCD4* were expressed at a higher level in ‘ZLE’ compared to ‘HJH’. We speculated that the low expression of *PmCCD1* and *PmCCD4* in ‘HJH’ caused the accumulation of carotenoids in the flowers, resulting in the appearance of yellow-flowered *P. mume*. Furthermore, *PmCCD4* showed a 2-fold and 22.5-fold higher expression in ‘ZLE’ than in ‘HJH’ during the S2 and S3 stages, respectively. To verify the expression levels of *PmCCDs* in different flowering stages of ‘HJH’ and ‘ZLE’, we carried out qRT-PCR analysis. The gene expression trends were consistent with the transcriptome data analysis (Figure 7B). Earlier studies have indicated that the variation in yellow flower color in chrysanthemums and petunias can be attributed to the cleavage of carotenoids facilitated by *CCD* genes [18,42]. We speculated that *PmCCD1* and *PmCCD4* might play a critical role, especially *PmCCD4*, in the formation of yellow flowers in ‘HJH’.

### 2.7. Expression Analysis of PmCCD Genes in Abiotic Stress Treatments

To investigate the ability of *PmCCD* genes in stress response, we conducted qRT-PCR analysis to assess the gene expression level under NaCl or PEG4000 treatments. Figure 8 displays that NaCl or PEG stress affected *PmCCD* gene expression levels. Under NaCl treatment, the expression of *PmCCD1*, *PmCCD4*, *PmCCD8* and *PmNCED6* increased and then decreased. In contrast, *PmCCD1* peaked at 3 h, while *PmCCD4*, *PmCCD8* and *PmNCED6* were most abundant at 6 h. *PmNCED3* and *PmNCED5*, belonging to the *NCED* subfamily, had down-expression in the early stage of NaCl treatment, and at 24 h, *PmNCED3* rapidly increased, while the expression of *PmNCED5* began to recover. The expression patterns of *PmCCD1-like-a* and *PmCCD1-like-c* in the *CCD-like* subfamily were similar, with the highest accumulation at 12 h, and then decreased rapidly. Only *PmCCD7* expressed a downward trend, without expression at 3 h to 24 h (Figure 8A). Under PEG treatment, *PmCCD* genes showed a new expression pattern. *PmCCD1* and *PmCCD4* possessed high expression at 24 h, indicating that *PmCCD1* and *PmCCD4* might have the potential to resist PEG stress persistently. The expression accumulation of *PmCCD8*, *PmNCED3* and *PmCCD1-like-c* was the highest at 6 h. However, the expression levels of *PmNCED5* and *PmNCED6* decreased during 1–24 h. *PmCCD7* was only expressed at 1 h, which was 14.8 times higher than that at 0 h, and then the expression decreased rapidly at 3–24 h (Figure 8B). *PmCCD1-like-b* was expressed in neither NaCl nor PEG treatment. In conclusion, *PmCCD* genes might participate in NaCl and PEG stress responses, and each *PmCCD* gene had a unique expression pattern.

### 2.8. Protein Interaction Network Analysis

Interaction network analysis could uncover the relationship between proteins and predicted protein function. Using the AraNet V2 website, we constructed an interaction network with a homologous protein from *Arabidopsis*. Figure 9 displays seven PmCCD proteins with orthologous proteins in *Arabidopsis*, and CCD family members interacted with each other: CCD1 and CCD8, CCD7 and NCED3, CCD7 and CCD1, CCD7 and CCD8, NCED5 and NCED6, NCED5 and CCD1 and NCED6 and CCD1. This suggested CCD members might participate in the same signal transduction and biological processes by forming dimers or polymers through protein interactions.

The interaction network diagram showed that CCDs could interact with proteins, such as MYB101, RD26, ATAF1, CHY1, ZOS, SnRK3.6, VAR3 and SAT, suggesting CCDs might function through interactions with other proteins. Further analysis revealed that CCDs had 142 interacting proteins, of which CCD1 protein had the most interacting proteins with 33, while CCD7 protein had only 12 interacting proteins. The interaction network displayed that CCDs were involved in essential roles in the biosynthesis and signal transduction of hormones such as ABA, GA and auxin. CCDs functioned in various plant growth and development processes, such as embryo and stamen development. They also responded to biotic stresses like pathogen invasion and abiotic stresses, including drought, low temperature, high temperature, injury and high salinity. CCDs were involved in metabolic processes, such as carotenoid metabolism and chlorophyll biosynthesis. PmCCD family proteins might have similar functionality to AtCCD. Overall, the interaction network served as a valuable reference for investigating the potential functions of *PmCCDs*.

### 2.9. Interaction of PmCCD4 with Other Proeins

We utilized the Uniprot website to further predict the interaction between PmCCD4 and other proteins. The website showed that CCD4 could interact with VAR3 (variegated 3) and SAT (serine acetyltransferase) to perform its function in *Arabidopsis*. In Figure 9, the interaction network also predicted interactions between PmCCD4 and PmVAR3 or PmSAT. Therefore, the interaction between PmCCD4 and PmVAR3-1/2 or PmSAT3/5 was validated using yeast two-hybrid assays and luciferase complementation experiments. The yeast two-hybrid assays showed that the combinations of pGBKT7-PmCCD4 and pGADT7-PmVAR3-1/2, as well as pGBKT7-PmCCD4 and pGADT7- PmSAT3/5, were able to grow and turn blue on an SD/-Leu/-Trp/-His/-Ade/x-α-gal solid medium (Figure 10A). This indicated that PmCCD4 could interact with PmVAR3-1/2 and PmSAT3/5 in vitro. Next, the detection of a strong chemiluminescent signal in the luciferase complementation test group confirmed the interaction of PmCCD4 with Pm VAR3-1/2 or PmSAT3/5 separately. Moreover, the high luciferase activity also supported this conclusion (Figure 10B).

## 3. Discussion

CCD, a small gene family, was involved in the forming of aromatic compounds and plant hormones from catalytic carotenoids, contributing to the fragrance, color, abscisic acid and strigolactone formation in plants. Currently, the CCD family genes have been identified in various plants, such as 9 in *Arabidopsis* [43], 19 in tobacco [44], 7 in tomato [9], 21 in apple [10], 30 in rapeseed [45] and 12 in strawberry [46]. In this research, we identified 10 *PmCCD* genes through the *P. mume* genome. Multiple sequence alignment revealed that all the proteins exhibited the conserved RPE65 domain. Additionally, most of the *PmCCD* genes contained conserved active sites with His residues. The conserved His residues in CCD proteins determined their enzymatic activity [47]. We speculated that the majority of PmCCD proteins possessed catalytic activity. The 10 *PmCCD* genes were distributed unevenly among chromosomes 2, 3 and 5. The *CCD* gene family might have specific evolutionary patterns across different species. Subcellular localization predictions suggested PmCCD1 and PmCCD1-like-c were likely to function in the cytoplasm, indicating their potential exclusion from chlorophyll photosynthesis. CCD4 was located in plastids to cleave carotenoids. We hypothesized that PmCCD4 exhibited similar catalytic activity. The localizations of PmNCED3, PmNCED5 and PmNCED6 in the chloroplast were consistent with previous studies [48]. In *Arabidopsis*, the collaboration of AtCCD7 and AtCCD8 converted β-carotene into the caprolactone belonging to strigolactone precursor, subsequently enhancing plant growth and development [49]. The chloroplast localizations of PmCCD7 and PmCCD8 were intimately linked to the synthesis pathway of strigolactone, which was intricately involved in the cleavage of β-carotene in plastids.

The CCD gene family in the majority of plants was classified into two major subfamilies: CCD and NCED. In *Arabidopsis*, the CCD subfamily contained four members: CCD1, CCD4, CCD7 and CCD8. The NCED subfamily included five members, namely NCED2, NCED3, NCED5, NCED6 and NCED9. Furthermore, a novel group called CCD-like (CCDL) have been identified in many species [8,44]. Homologs of CCDL genes were absent in *Arabidopsis*. According to the phylogenetic tree analysis, we characterized and categorized the *PmCCD* gene family into CCD1, CCD4, CCD7, CCD8, NCED (NCED3, NCED5 and NCED6) and CCD-like subfamilies. The number and grouping of CCD family members in *P. armeniaca* and *P. persica* were consistent with PmCCD, and PmCCD proteins exhibited a closer genetic distance to ParCCD. Compared to *Arabidopsis*, the *PmCCD* gene family lacked *NCED2* and *NCED9*, which suggested a gradual functional replacement during the evolutionary process of *PmNCED* genes. However, the *PmCCD* gene family possessed an additional CCDL subfamily. In addition, the NCED subfamily was most closely related to the CCD1 and CCD4 groups, while their relationship with the CCD7 and CCD8 groups was the furthest.

Genomic structure and motif analysis revealed considerable variations in exon/intron patterns and conserved motifs among *PmCCD* members across different subfamilies. Notably, within the same subfamily, a general similarity was observed in terms of the quantity and distribution of exons/introns and conserved motifs. All members of the *PmNCED* subfamily lacked introns, aligning with previous reports in plants [7,8]. Furthermore, PmNCED contained the highest number of conserved motifs, which were more conserved than other groups. This phenomenon was widespread in plants [50]. Based on motif analysis and sequence alignment, except for PmCCD1-like-b and PmCCD1-like-c, PmCCD proteins possessed the four conserved histidine residues required for enzymatic activity. Currently, it remains unclear whether the absence of conserved histidine residues in PmCCD1-like-b and PmCCD1-like-c would affect enzymatic activity and protein functionality. Therefore, elucidating the active sites will contribute to exploring the functionality of CCDL genes, potentially leading to new insights into the CCD gene family.

Gene duplication events have exerted a significant impact on the evolutionary trajectory of gene families [51]. These duplications not only expanded the gene family but also played a pivotal role in gene diversification, potentially leading to significant morphological changes in plants. Our intraspecific collinearity analysis of 10 *PmCCD* genes revealed the absence of intraspecific collinearity among *PmCCD* genes. This suggested that gene duplication events might not be the primary driver of *PmCCD* gene expansion. Interspecific collinearity analysis serves as a valuable approach for exploring the evolutionary dynamics of gene families across different species. We conducted interspecific collinearity analysis between *P. mume* and two other *Prunus* species (*P. armeniaca* and *P. persica*) and had homologous gene pairs. According to the Ka/Ks values, the CCD orthologous pairs showed an earlier divergence in *P. mume* and *P. armeniaca*. This observation provided additional evidence supporting a closer phylogenetic relationship between them.

The tissue-specific expression patterns of genes are closely associated with their functional characteristics. To investigate the expression patterns of the *PmCCDs* in various tissues of *P. mume*, we employed qRT-PCR technology in the seven tissues (stem, leaf, petal, fruit, sepal, stamen and pistil). The *PmCCD* genes exhibited distinct tissue specificity. Among them, *PmCCD1* and *PmCCD4* were expressed highly in petals. Research showed that *CCD1* and *CCD4* primarily participated in the form of volatile compounds and flower color through cleavage carotenoids [52,53,54,55]. We speculated that *PmCCD1* and *PmCCD4* might be involved in the formation of flower color and fragrance in the ‘HJH’. *CCD7* and *CCD8* were involved in the synthesis of strigolactone which was primarily synthesized in the roots [56]. *PmCCD7* was highly expressed in the root, consistent with *AtCCD7* [57]. This implied that *PmCCD7* might have similar functionality to *AtCCD7*. In addition, *PmCCD8* exhibited higher expression levels in stems and leaves, suggesting that *PmCCD8* primarily functions in stems and leaves. *PmNCED3*, *PmNCED5* and *PmNCED6* were highly expressed in stamens, fruits and leaves, respectively. This indicated that genes within the same family gradually acquired different functions during the evolutionary process. Members of the *PmCCD1-like* subfamily had stem-specific expression while showing low expression in other tissues. The *PmCCD1-like* genes were likely to have unique functions in stems.

*CCD1* and *CCD4* participate in the degradation of carotenoids, providing unique colors, flavors and aromas to fruits and flowers. Previous studies have shown that *CCD1* regulates the volatile aroma compound β-ionone production in petunia, tomato and *L. chinense* [14,17,58]. The variation in petal color among different chrysanthemum mutants was attributed to differential expression levels of the *CmCCD4* gene, resulting in variations in carotenoid content [59]. Inhibiting the expression of the *CmCCD4* gene through RNAi could transform white chrysanthemum flowers into yellow flowers [18]. In *Brassica* species, mutation of the *CCD4* genes led to the formation of yellow flowers [20]. Similarly, in *O. fragrans* and *Rhododendron japonicum*, the expression level of the *CCD4* gene determined the carotenoid content, thereby leading to differences in flower color [60,61]. The *CCD4* gene changed *Lilium brownie* petals from yellow to white one day after anthesis [62]. Compared to other tissues, *PmCCD1* and *PmCCD4* were expressed highly in petals. Next, we analyzed the expression of *PmCCD* genes during the flowering process of the yellow-flowered ‘HJH’ and white-flowered ‘ZLE’. The results revealed that the expression levels of *PmCCD1* and *PmCCD4* in the yellow-flowered ‘HJH’-S2 and S3 were lower than those of the white-flowered ‘ZLE’. We speculated that the low expression of *PmCCD1* and *PmCCD4* in ‘HJH’ caused the accumulation of carotenoids in the flowers, resulting in the appearance of yellow-flowered *P. mume*. Whether *PmCCD1* and *PmCCD4* are related to the formation of floral fragrance requires further exploration.

In plants, *CCD* genes can participate in abiotic stress response, while the involvement of *PmCCDs* in abiotic stress has not been studied so far. The analysis of the *PmCCDs* promoter showed that the *PmCCDs* promoter region contained several cis-acting elements related to stress, such as W-box, MYB, LTR, MYC, etc. Therefore, the expression pattern of *PmCCDs* treated with NaCl and PEG4000 was investigated by qRT-PCR. We found that *PmCCD1* and *PmCCD4* were involved in response to NaCl and PEG treatments. *PmCCD7* could respond to PEG stress within 1 h and was not persistent. Additionally, *PmCCD8* displayed responsiveness to NaCl stress and PEG stress. Based on the available information, *PmCCD8* might have a higher potential for stress resistance than *PmCCD7*. However, it is essential to conduct further experiments and investigations to fully understand and validate the comparative stress resistance potential between *PmCCD7* and *PmCCD8*. The NCED subfamily was involved in plant resistance by regulating ABA synthesis [63,64]. The biosynthesis of ABA in *Arabidopsis* involved the participation of *AtNCED3*, *AtNCED5*, *AtNCED6* and *AtNCED9* [30,65]. The promoters of *PmNCED* genes were enriched with ABRE motifs, indicating that the *PmNCED* subfamily was involved in ABA-related biological processes of *P. mume*. Under NaCl treatment, the expression abundance of *PmNCED3* and *PmNCED6* was the highest at 24 h and 6 h, respectively. However, only *PmNCED3* was expressed in PEG treatment. It is worth noting that *PmNCED3* may play a more significant role in osmotic stress, which requires further validation. Regarding the *CCD-like* group, the functional characterization of these genes is yet to be investigated. *PmCCD1-like-a* and *PmCCD1-like-c* had the potential to resist stress. Exploring the functionality of CCD-like group genes will provide a new perspective for studying the *CCD* gene family.

Interacting proteins might regulate each other, be closely related in function or participate in the same signaling pathway or physiological process. According to yeast two-hybrid system and luciferase complementation experiments, PmCCD4 interacted with PmVAR3 protein or PmSAT3/5 proteins. In *Arabidopsis*, AtVAR3 interacted with AtCCD4, and VAR3 could regulate the function of the CCD4 enzyme as a signaling molecule, thereby regulating carotenoid catabolism [66]. In *P. mume*, whether the binding of PmVAR3 protein with PmCCD4 affects the function of PmCCD4 to cleavage carotenoid still needs further exploration. SAT genes were involved in plant development and response to various stresses [67,68]. We speculated that PmSAT3/5 might have stronger resistance to various stresses when interacting with PmCCD4 protein. This is also a topic worthy of further study.

## 4. Materials and Methods

### 4.1. Identification of PmCCDs

The genome of *P. mume* was available on NCBI database [69]. Using Pfam database, we downloaded the configuration file for the REP65 domain (PF03055). The HMMER3 software was used to obtain the CCD protein of *P. mume* with E-value ≤ 10^−5^ [41]. The REP65 domains of the PmCCD proteins were determined using NCBI-CD-search.

The ExPASy (https://web.expasy.org/protparam/, accessed on 15 June 2023) was employed to seek CDS length, theoretical isoelectric points (pI), amino acid sequences (aa), instability index, molecular weights (MWs) and aliphatic index grand averages of hydropathicity (GRAVYs) of PmCCDs. The N-terminal targeting signals and subcellular localization were predicted through iPSOR (http://ipsort.hgc.jp/, accessed on 15 June 2023) and WoLF PSORT (https://wolfpsort.hgc.jp/, accessed on 16 June 2023), respectively [70]. The SOPMA website (https://npsa-prabi.ibcp.fr/cgi-bin/npsa_automat.pl?page=npsa_sopma.html, accessed on 16 June 2023) was utilized for protein secondary structure prediction [71].

### 4.2. Phylogenetic Tree Analysis and Protein Sequence Alignment of PmCCD Proteins

To unravel the evolution of PmCCD proteins, we constructed an evolutionary tree using CCD gene families from *P. armeniaca* and *P. persica*, as well as *Arabidopsis*. The genomes of *P. armeniaca* and *P. persica* were acquired from NCBI, and the sequences of *AtCCD* were sourced from TAIR database (https://www.arabidopsis.org/, accessed on 17 June 2023). MEGA7.0 was utilized to construct the phylogenetic tree (bootstrap = 1000) with the maximum likelihood method. The phylogenetic tree was drawn by using iTOL (https://itol.embl.de/, accessed on 17 June 2023). Multiple sequence alignment of 10 PmCCD gene family members was performed using MEGA7.0 and visualization through Muscle (https://www.ebi.ac.uk/Tools/msa/muscle/, accessed on 17 June 2023).

### 4.3. Gene Structure and Conservation Motif Analysis of PmCCD Sequences

The structure and conserved domain of *PmCCD* genes were estimated and drawn by using TBtools [72]. The motif structure of PmCCD proteins was explored through the MEME program (the motif limit was set to 15), and other default parameters were used.

### 4.4. Chromosome Localization and Collinearity Analysis of PmCCDs

According to the chromosome location information of *PmCCD* genes provided by *P. mume* genome database, we mapped the gene distribution with MG2C tool (http://mg2c.iask.in/mg2c_v2.1/, accessed on 17 June 2023). Intraspecific and interspecific collinearity of *CCD* genes (*P. mume*, *P. armeniaca* and *P. persica*) were analyzed and visualized using MCScanX [73]. The mutation rates of nonsynonymous replacement rate (Ka) and synonymous replacement rate (Ks) of *CCD* collinear gene pairs were predicted by using TBtools. *PmCCD* gene divergence time (T) was calculated by the following equation: T = dS/2λ × 106 Mya, λ = 1.5 × 10^−8^ for dicots [74].

### 4.5. Analysis of PmCCDs Expression Pattern

Expression patterns of *PmCCDs* in different varieties and various tissues: The 5-year-old *P. mume* ‘Huangjinhe’ (‘HJH’) with yellow flowers and 5-year-old *P. mume* ‘Zaolve’ (‘ZLE’) with white flowers were planted in the natural environment of Chongqing, China. The ‘HJH’ is a rare yellow flower variety in *P. mume*. However, the yellow color of the ‘HJH’ is not stable. As the flowers blossom, the yellow color gradually fades away. During the big bud stage (Stage2, S2), the petals were yellow. In the half-bloom stage (Stage3, S3), the outer edge of the petals turned white. The petals were almost white during the full-bloom stage (Stage4, S4). To explore the role of *PmCCDs* in *P. mume* with yellow flowers, the petals of S2, S3 and S4 were taken from ‘HJH’ and ‘ZLE’ for transcriptome sequencing (NCBI accession no. PRJNA854285). Seven tissues, including stems, leaves, petals, fruits, sepals, stamens and pistils were collected from the ‘HJH’. Meanwhile, TBtools was used to plot expression heatmaps of 10 *PmCCDs* at flower buds, fruits, leaves, roots and stems of *P. mume* wild species (GEO No. GSE40162), as well as three flowering stages of ‘HJH’ and ‘ZLE’.

Expression patterns of *PmCCDs* under abiotic stress treatments: The annual stems (20 cm) of ‘ZLE’ with the same growth state were obtained in April. The annual stems were treated by inserting them into 160 mmol/L NaCl and 20%PEG4000 solution, separately. Samples were taken at 0 h, 1 h, 3 h, 6 h, 12 h and 24 h, respectively. Using the untreated annual stems (0 h) as a blank control, qRT-PCR was performed on the treated annual stems. The plant samples were subjected to identical air humidity (65–70%), light duration (16 h/8 h) and light intensity (150 μmol·m^−2^·s^−1^). The samples were stored at −80 °C to maintain their integrity and stability until RNA extraction, with 3 biological replicates.

### 4.6. RNA Extraction and qRT-PCR Analysis

Total RNA from various tissues and varieties of ‘HJH’ was isolated according to the user manual of the RNA extraction kit (Takara, Beijing, China). The synthesis of first strand cDNA by RNA (1 µg) was reversed with PrimeScriptTMRT Reagent Kit with gDNA Eraser reagent (Takara, Beijing, China). TB Green II Premix Ex Taq (Takara, Beijing, China) was employed to perform a qRT-PCR system (10 µL), including 2 µL template cDNA. The relative expression levels were calculated by using 2^−ΔΔCt^ method, with the reference gene being the protein phosphatase 2A (PP2A) gene. Ten specific primers for *PmCCDs* are shown in Appendix A. A minimum of three replicates were performed for each qRT-PCR assay.

### 4.7. Analysis of PmCCD Promoter Element and Protein Interaction Network

From the genome of *P. mume*, we searched for the *PmCCD* sequences and extracted the 2000 bp upstream of the TSS (Transcription Start Site) as the promoter. The promoter cis-acting regulatory elements were operated using PLACE (https://www.dna.affrc.go.jp/PLACE/?action=newplace, accessed on 17 June 2023) [75]. The AraNet V2 tool was used to construct protein interaction networks of PmCCD proteins [76]. The visualization of the protein interaction network was performed using Cytoscape and STRING software (http://string-db.org/, accessed on 17 June 2023) [77].

### 4.8. Yeast Two-Hybrid System

*PmCCD4* was constructed into *pGBKT7* vector, and *PmVAR3-1/2* and *PmSAT3/5* genes were constructed into *pGADT7* vector. Plasmid *pGBKT7-PmCCD4* and *pGBKT7* were transferred into *Y2HGold* yeast strains, coated on SD-trp/X-α-gal defect screening plates and cultured at 30 °C for 3–5 days for yeast self-activation detection. Then, the *pGBKT7-PmCCD4* and *pGADT7-PmVAR3-1/2* or *pGADT7-PmSAT3/5* genes were co-transferred to *Y2HGold* strain and coated on SD-trp-his and SD-trp-leu-his-ade/X-α-gal defect screening medium, respectively. The proteins were observed and determined for interactions. Specific primers are listed in Appendix A.

### 4.9. Luciferase Complementation Experiment

*PmCCD4* was linked to the *nLUC* vector, and the *PmVAR3-1/2* and *PmSAT3/5* genes were linked to the *cLUC* vector. The recombinant vectors were transformed into an *Agrobacterium* strain of GV3101 (pSoup-p19) and expressed in *N. benthamiana*. We applied the prepared 0.32 mg/mL D-luciferin potassium salt solution to the leaves. Then, we observed and took photos using the molecular imaging system LB983 NightOwl II (Berthold Technologies, Baden-Württemberg, Germany).

## 5. Conclusions

In this study, 10 non-redundant *PmCCD* genes were identified in *P. mume*. The characteristics of *PmCCD* genes showed diversity. All PmCCD proteins had the RPE65 conserved domain. In addition, most CCD proteins contained four conserved histidine sites. Based on the phylogenetic tree, PmCCD proteins were divided into six subfamilies: CCD1, CCD4, CCD7, CCD8, NCED and CCD-like. *PmCCD* genes exhibited interspecific collinearity with *P. armeniaca* and *P. persica*. The analysis of promoter and protein interaction indicated that environmental factors and hormone signals had complex regulation of *PmCCD* genes. The qRT-PCR results showed that *PmCCD* genes had obvious tissues and varieties. Compared with white-flowered ‘ZLE’, *PmCCD1* and *PmCCD4* genes were low-expressed in ‘HJH’ with yellow petals, which suggested *PmCCD1* and *PmCCD4* might be related to the formation of yellow flowers in *P. mume*. Nine *PmCCD* genes could respond to NaCl or PEG treatment. These genes might play a crucial role in salt and drought resistance in *P. mume*. Moreover, PmVAR3 and PmSAT3/5 were found to interact with PmCCD4 protein in yeast and tobacco leaf cells. These results will establish valuable bioinformatics foundations for further exploring the functions of *PmCCD* genes.

## Figures and Tables

**Figure 1 ijms-24-13950-f001:**
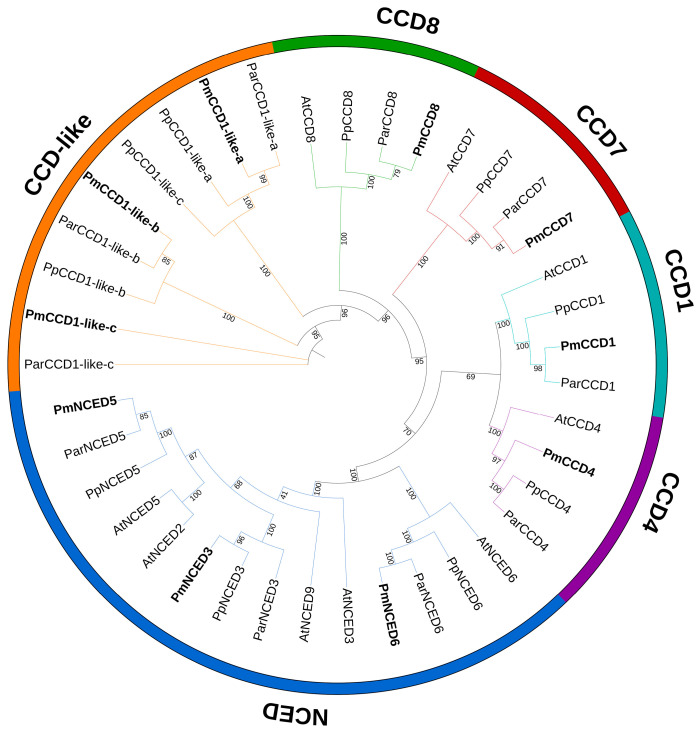
Phylogenetic tree of CCD proteins of *P. mume* and other plant species. A maximum likelihood (ML) tree with 1000 bootstrap replicates was constructed using MEGA7. Turquoise denotes CCD1 subfamily, purple denotes CCD4 subfamily, red denotes CCD7 subfamily, green denotes CCD8 subfamily, blue denotes NCED subfamily, and orange denotes CCD-like subfamily.

**Figure 2 ijms-24-13950-f002:**
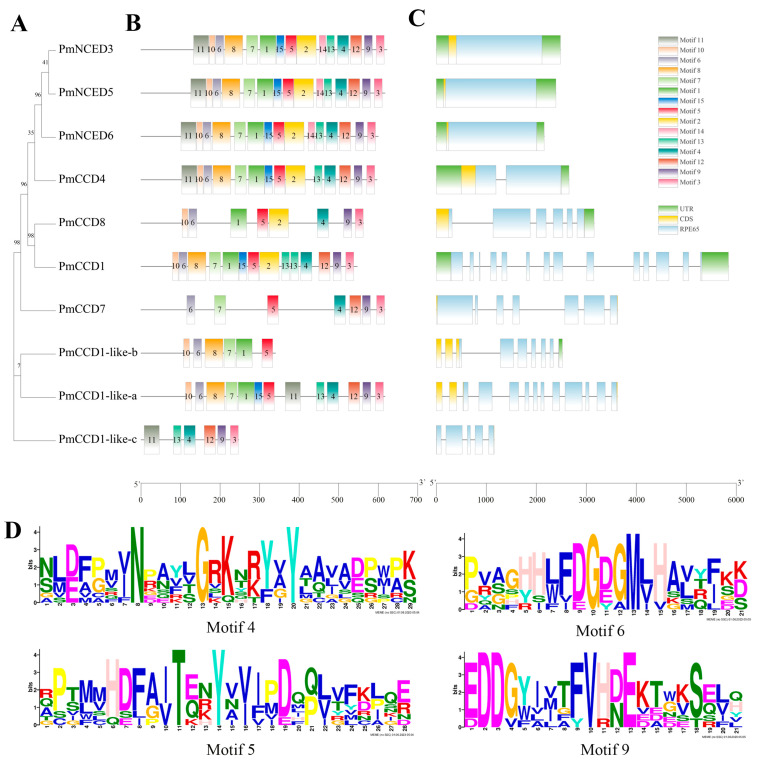
Phylogenetic relationship, motif analysis and gene structure of CCD gene family in *P. mume*. (**A**) Phylogenetic tree of PmCCD proteins. (**B**) MEME was utilized to analyze the motif composition of PmCCD proteins. Different colored rectangles with numbers 1–15 represent distinct patterns. (**C**) Gene structure of *PmCCDs*. Introns are visualized using gray lines. Green boxes indicate UTR regions, yellow boxes represent the CDS regions, and blue boxes denote the RPE65 conserved domain of PmCCDs. (**D**) Protein sequences of motifs 4, 5, 6 and 9.

**Figure 3 ijms-24-13950-f003:**
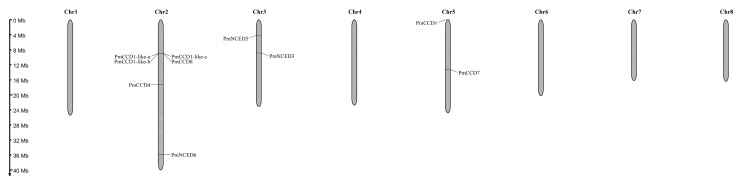
Chromosome distribution of the *PmCCD* genes.

**Figure 4 ijms-24-13950-f004:**
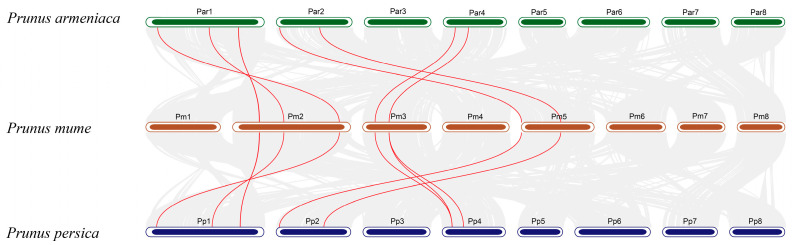
Analysis of collinearity of *CCD* genes in *P. mume*, *P. armeniaca* and *P. persica*. The red lines represent the collinear gene pairs within the *CCD* gene family, while the gray lines represent other collinear gene pairs within the genome. Red indicates the chromosomes of *P. mume*, green means the chromosomes of *P. armeniaca*, and blue represents the chromosomes of *P. persica*.

**Figure 5 ijms-24-13950-f005:**
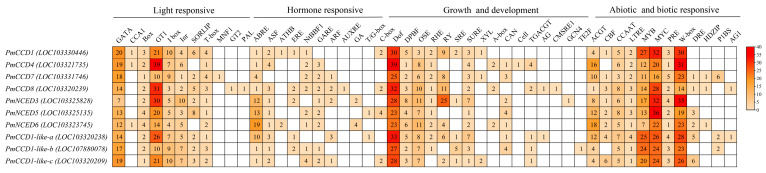
The cis-element analysis of *PmCCD* gene promoter regions. The heatmap shows different types of cis-elements in the *PmCCD* genes. Elements were divided into four categories. The colored scale refers to the number of each cis-element in the promoter of the *PmCCD* genes. Numbers represent the number of cis-elements in a gene.

**Figure 6 ijms-24-13950-f006:**
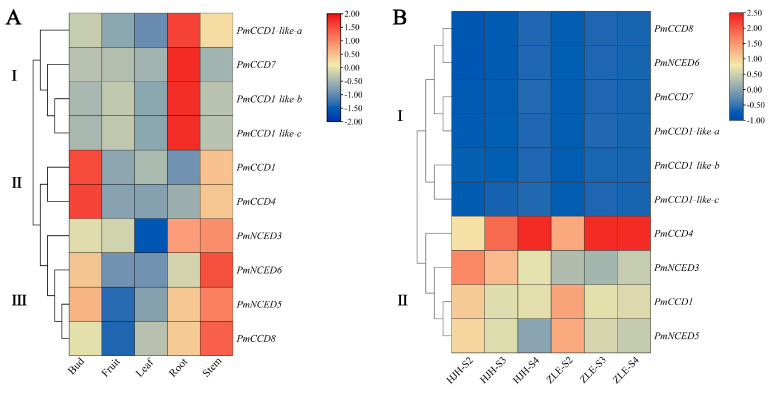
Hierarchical clustering heatmap of *PmCCD* gene expression profiles in various tissues and varieties. (**A**) Heatmap of *PmCCD* gene expression profiles in various tissues. (**B**) Heatmap of *PmCCD* gene expression profiles in ‘HJH’ and ‘ZLE’. Roman numerals mean classification. The colored scale represents the relative expression levels and is displayed at the right. Red means high expression, while blue means low expression.

**Figure 7 ijms-24-13950-f007:**
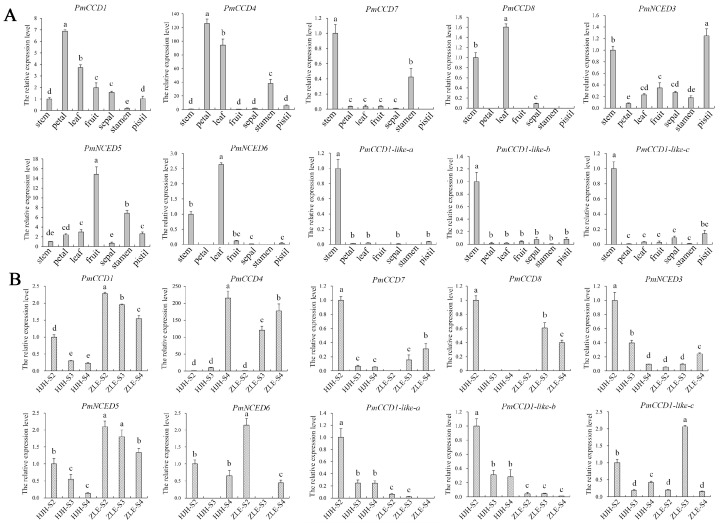
qRT-PCR analysis of the expression patterns of 10 *PmCCD* genes in various tissues and varieties. (**A**) qRT-PCR analysis of *PmCCD* genes in different tissues (stem, leaf, flower, fruit, sepal, stamen and pistil) of the ‘HJH’ variety. The x-axis represents the tissue types, while the y-axis indicates the relative expression levels. (**B**) qRT-PCR analysis of *PmCCD* gene expression in ‘HJH’ and ‘ZLE’. The x-axis represents the different flowering stages of ‘HJH’ and ‘ZLE’. The relative expression levels are depicted on the y-axis. Standard deviation error bars represent three independent replicates. Statistically significant differences (*p* < 0.05) are denoted by different letters (a–e).

**Figure 8 ijms-24-13950-f008:**
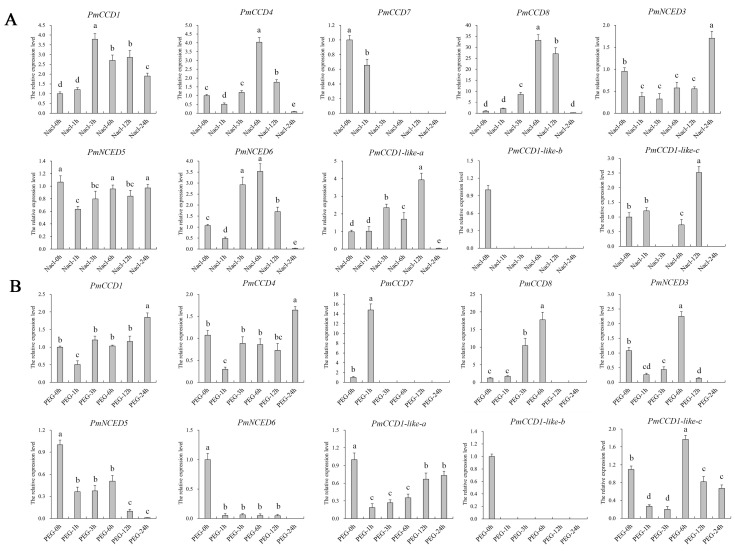
The qRT-PCR analysis of the expression patterns of *PmCCD* genes under stress treatment. (**A**) Expression patterns of *PmCCD* genes under NaCl treatment. (**B**) Expression patterns of *PmCCD* genes in PEG4000 treatment. The x-axis represents stress treatment time. The relative expression levels are depicted on the y-axis. Standard deviation error bars represent three independent replicates. Statistically significant differences (*p* < 0.05) are denoted by different letters (a–e).

**Figure 9 ijms-24-13950-f009:**
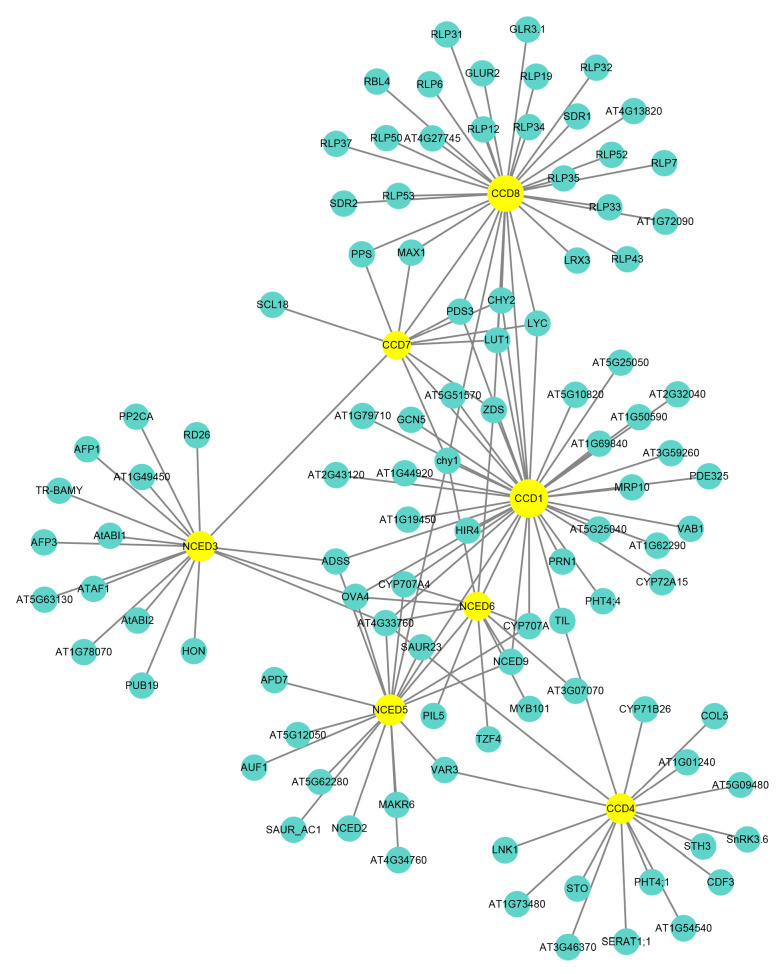
Analysis of PmCCD protein interaction network in *P. mume*. The yellow circle represents PmCCD proteins, and the blue circle represents interacting proteins.

**Figure 10 ijms-24-13950-f010:**
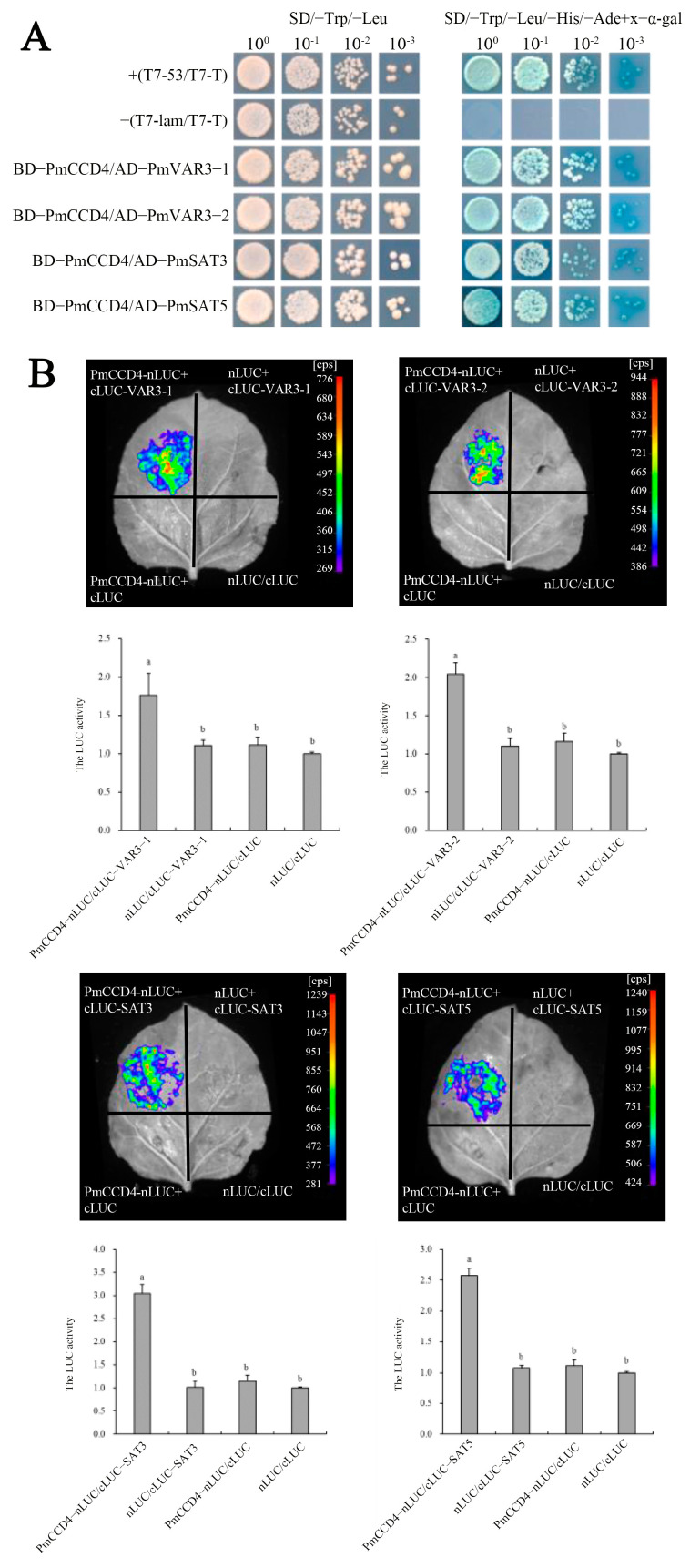
Interaction of PmCCD4 with PmVAR3 or PmSAT3/5 by using yeast two-hybrid system and luciferase complementation experiment. (**A**) Yeast two-hybrid assay for protein–protein interaction between PmCCD4 and PmVAR3 or PmSAT3/5. T7-53/T7-T means a positive control; T7-lam/T7-T means a negative control. BD and AD represent the pGBKT7 and pGADT7 vectors, respectively. The SD/-Trp-Leu medium lacks tryptophan and leucine. The SD/-Trp-Leu-His-Ade medium lacks tryptophan, leucine, histidine and adenine. X-α-gal was used to stain positive colonies. (**B**) The interaction between PmCCD4 and PmVAR3 or PmSAT3/5 was determined by using luciferase complementation experiment. Different regions of *Nicotiana benthamiana* leaves were injected with *Agrobacterium* GV3101 (pSoup-p19) strains carrying different fusion vectors. Fluorescence imaging was conducted 3 days after injection. Scale bar, 1 cm. cps, signal counts per second. Luminescence activity was measured. Standard deviation error bars represent three independent replicates. Statistically significant differences (*p* < 0.05) are denoted by different letters (a–b).

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
