# Peer review of "Phylogeny of PmCCD Gene Family and Expression Analysis of Flower Coloration and Stress Response in Prunus mume"

_ijms, 2023, doi:10.3390/ijms241813950_

Round 1

Reviewer 1 Report

The authors studied the phylogeny of the PmCCD gene family and the role of PmCCD gene family in flower coloration and stress response in Prunus mume. The study is relevant, well-structured, the methods used are suitable, and the results are well presented. In general, the obtained results have fundamental value and will be of interest to readers.

The paper can be accepted for publication in International Journal of Molecular Sciences.

Author Response

Thank you for your careful guidance.

Reviewer 2 Report

The manuscript entitled” Phylogeny of PmCCD gene family and expression analysis in flower coloration and stress response in Prunus mume” by Ding et al. aimed to investigate the role of   PmCCD gene family in ornamental woody Prunus mume. The authors have tried their best to support their findings.

However, I have the following major comments for the authors to improve the manuscript:

1.   Introduction needs revision. Since the title stresses on the role of PmCCD gene family in response to both flower coloration and stress responses, the authors need to expand the role of CCD genes in response to different stress conditions. They have only mentioned about it in two lines and the reference they have cited is quite old. Please write a paragraph regarding the role of CCD genes in response to different stress conditions by giving recent references.

2.  In the Materials and methods, under section 4.7. Analysis of PmCCD Promoter Element and Protein Interaction Network, the authors have not indicated regarding the length of the promoter sequences, extraction etc. Please give a detailed description regarding the length of the promoter sequences that you have used for the analysis of cis-elements. Please also provide the information regarding the extraction of promoter sequences i.e. from which Site/point you have extracted the genomic DNA as it is important to extract the right sequences.

   In the Materials and Methods, under section 4.7. Analysis of PmCCD Promoter Element and Protein Interaction Network, the authors have mentioned, “The promoter cis-acting regulatory elements were operated by PlantCARE (http://bioinformatics.psb.ugent.be/webtools/plantcare/html/) [56].”

    The authors have used Plant CARE database for promoter cis-element  analysis. However, this database is very old and does not have the up to date information. Hence, you are certainly going to miss many important and new cis-elements. The best option is to use the MATCH program in TRANSFAC (geneXplain) which you need to pay for the subscription. However, the authors can also use PlantPAN and PLACE database if you do not have access to TRANSFAC.

3.      Please provide the nucleotide numbering, 5’, and 3’ position of the promoter sequences in Figure 5.

4.      Please indicate A and B in Figure 6.

5.      Please improve the quality of Figure 7. It is very hard to read it.

6.   In the Materials and Methods, under the section 4.5. Analysis of PmCCDs Expression Pattern, please mention regarding the age of the P. mume 'Huangjinhe' ('HJH') with yellow flower and P. mume 'Zaolve' ('ZLE') with white flower.  Please provide a detailed description of the dataset, sequencing and data processing.

7.      Please provide the transcriptomics data for all treatments.

8.    Discussion also needs revision. Please improve the role and mechanism of PmCCD genes in different tissues, response to yellow coloration, stress response, and discuss them systematically and separately.

9.      Did you find any cis-elements associated with different transcription factors involved in various stress response mechanism in the promoter of PmCCD genes? It will give you a lot of information regarding the stress response regulatory mechanism by these genes.

10.  Please refine the language.

Minor editing is required.

Author Response

Please see the attachment,Thank you for your careful guidance.

Reviewer 3 Report

Dear authors,

I would have some suggestions to improve the quality of the manuscript. Here is my suggestions/quarries:

1.       What kind of stress conditions, you have used in your study?

2.       How did authors come to conclusions that flower colorations are related with gene expression without targeting the right target/gene i. e. functional analysis?

3.       How authors identified the CCD genes family ? and  what is their criteria to validate in qPCR?

4.       Fig. 7. is very short and blur, it should be changed.

5.       I suggest to authors should change their title of the manuscript and  conclusion as they not confirmed the right candidate genes related with flower coloration

Author Response

(The authors gave the same response as above.)

Round 2

Reviewer 2 Report

Thank you authors for the revision.

However, I have the following major comment for the authors.

1.      In the Materials and Methods, under section 4.7. Analysis of PmCCD Promoter Element and Protein Interaction Network, line 593-596, the authors have mentioned, “From the genome of P. mume, we searched for the PmCCD sequences and extracted  the 2000 bp upstream of the start codon ATG as the promoter.”

The authors have taken 2000 bp upstream of the start codon (ATG) site, which is Translation Initiation Site (TIS). The sequences they have taken could be a part of Exon 1 and a part of promoter or only Exon 1 depending on the gene length. So, the cis-element detected in those regions are not correct, as the sequences are not right promoter sequences. Sometime, some distal cis-elements are present in the downstream of promoter sequence, but majority are present in the upstream part of promoter sequence. Hence, please extract the promoter sequences from the upstream of   TSS (Transcription Start Site) and redo the cis-element enrichment analysis.

2.      Please revise the method, results and discussion part based on your new promoter cis-element analysis.

3.      Please check the minor language errors throughout the manuscript.

Minor language editing is required. 

Author Response

Please see the attachment,thank you for the correction and careful guidance.

Reviewer 3 Report

Now, the authors compile all the quarries and questions, therefore, manuscript can be accepted for publication. I congratulate authors for their work.

Author Response

Thank you for the correction and careful guidance.

Round 3

Reviewer 2 Report

Thank you authors for revising the manuscript.

However,  there is still mistake regarding the extraction of promoter sequences in the manuscript. 

Regarding the extraction of promoter sequences, the authors replied, "Thank you for the correction and careful guidance. We have extracted

the promoter sequences from the upstream of TSS (Transcription Start Site) and redo the cis element enrichment analysis using the PLACE database."

However, in the Materials and methods, under the section 4.7. Analysis of PmCCD Promoter Element and Protein Interaction Network, line 593-595, the authors mentioned, "From the genome of P. mume, we searched for the PmCCD sequences and extracted  the 2000 bp upstream of the start codon ATG as the promoter. The promoter cis-acting regulatory elements were operated by PlACE (https://www.dna.affrc.go.jp/PLACE/?ac-595 tionnewplace) [75]."

I am not sure whether the authors have the proper understanding regarding the promoter sequences? In my previous comment, I clearly mentioned regarding the exact location of the promoter sequences. 

You must revise the sentence regarding the extraction of promoter sequences before it gets accepted.

Also, please correct 'PlACE' to 'PLACE' database. 

Minor editing is required.

Author Response

Please see the attachment and thank you for the correction and careful guidance.
